# Trends and outcomes of percutaneous coronary intervention during the COVID-19 pandemic in Michigan

**Lorenzo Azzalini**[1], **Milan Seth**[2], **Devraj Sukul**[2], **Javier A. Valle**[3,4], **Edouard Daher**[5], **Brett Wanamaker**[2], **Michael T. Tucciarone**[6], **Anwar Zaitoun**[7], **Ryan D. Madder**[8], **Hitinder S. Gurm**[2]*

1 Division of Cardiology, VCU Health Pauley Heart Center, Virginia Commonwealth University, Richmond, VA, United States of America, 2 Department of Internal Medicine, Division of Cardiovascular Medicine, University of Michigan, Ann Arbor, MI, United States of America, 3 Michigan Heart and Vascular, Ann Arbor, MI, United States of America, 4 University of Colorado School of Medicine, Aurora, CO, United States of America, 5 Cardiac Catheterization Laboratory, Ascension St John Hospital, Detroit, MI, United States of America, 6 Beaumont Hospital, Troy, MI, United States of America, 7 Covenant Cardiology, Saginaw, MI, United States of America, 8 Spectrum Health Hospitals Fred and Lena Meijer Heart Center, Grand Rapids, MI, United States of America

* hgurm@med.umich.edu

## Abstract

### Background

The COVID-19 pandemic has severely impacted healthcare delivery and patient outcomes globally.

### Aims

We aimed to evaluate the influence of the COVID-19 pandemic on the temporal trends and outcomes of patients undergoing percutaneous coronary intervention (PCI) in Michigan.

### Methods

We compared all patients undergoing PCI in the BMC2 Registry between March and December 2020 ("pandemic cohort") with those undergoing PCI between March and December 2019 ("pre-pandemic cohort"). A risk-adjusted analysis of in-hospital outcomes was performed between the pre-pandemic and pandemic cohort. A subgroup analysis was performed comparing COVID-19 positive vs. negative patients during the pandemic.

### Results

There was a 15.2% reduction in overall PCI volume from the pre-pandemic (n = 25,737) to the pandemic cohort (n = 21,822), which was more pronounced for stable angina and non-ST-elevation acute coronary syndromes (ACS) presentations, and between February and May 2020. Patients in the two cohorts had similar clinical and procedural characteristics. Monthly mortality rates for primary PCI were generally higher in the pandemic period. There were no significant system delays in care between the cohorts. Risk-adjusted mortality was

**Data Availability Statement:** The authors are unable to share the raw data due to contractual agreements between participating institutions and the BMC2 registry that prohibit data sharing with

external agencies. However, the analysis code and metadata to support the study is available on request from Annemarie Forrest, Program Manager BMC2. (avassalo@med.umich.edu). The BMC2 registry is a collaborative of all non federal hospitals in Michigan. There are standing data use agreement with each hospital and the data use agreements limit sharing of raw data. The registry participants and collaborators can request analysis from the registry but the raw data are not shared outside (or even internally with the physician leaders/participants). Mr Seth as the statistician has access to all data, none of the other authors have access to raw data. Researchers from the participating hospitals (with external collaborators) can request analysis but such requests are not available to participants from outside the registry.

**Funding:** This work was supported by the Blue Cross Blue Shield of Michigan and Blue Care Network as part of the Blue Cross Blue Shield of Michigan Value Partnerships program. The funding source supported data collection at each site and funded the data-coordinating center, but had no role in study concept, interpretation of findings, or in the preparation, final approval or decision to submit the manuscript.

**Competing interests:** I have read the journal's policy and the authors of this manuscript have the following competing interests: Dr. Azzalini received consulting fees from Teleflex, Abiomed, Asahi Intecc, Abbott Vascular, Philips, GE Healthcare, and Cardiovascular Systems, Inc. Dr. Sukul receives salary support from the Blue Cross Blue Shield of Michigan for his role in BMC2. Dr. Gurm receives research support from Blue Cross and Blue Shield of Michigan, and Michigan Translational Research and Commercialization for Life Sciences Innovation Hub. He is the co-founder of, owns equity in, and is a consultant to Amplitude Vascular Systems. He also owns equity in Jiaxing Bossh Medical Technology Partnership and is a consultant for Osprey Medical. He is the chair of the Clinical Events Committee for the PERFORMANCE trial sponsored by Contego Medical. The other authors have no disclosures. This does not alter our adherence to PLOS ONE policies on sharing data and materials.

**Abbreviations:** AKI, acute kidney injury; CAD, coronary artery disease; CI, confidence interval; eGFR, estimated glomerular filtration rate; LVEF, left ventricular ejection fraction; NSTE-ACS, non-ST-elevation acute coronary syndrome; OR, odds ratio; PCI, percutaneous coronary intervention; SMD, standardized mean difference; STEMI, ST-elevation myocardial infarction.

higher in the pandemic cohort (aOR 1.26, 95% CI 1.07–1.47, p = 0.005), a finding that was only partially explained by worse outcomes in COVID-19 patients and was more pronounced in subjects with ACS. During the pandemic, COVID-19 positive patients suffered higher risk-adjusted mortality (aOR 5.69, 95% CI 2.54–12.74, p<0.001) compared with COVID negative patients.

## Conclusions

During the COVID-19 pandemic, we observed a reduction in PCI volumes and higher risk-adjusted mortality. COVID-19 positive patients experienced significantly worse outcomes.

## Introduction

The COVID-19 pandemic represents the most important public health crisis of the century. Besides its direct impact on COVID-19-related hospitalizations, morbidity and mortality, the pandemic has also dramatically impacted health care delivery for non-COVID-19 conditions around the world [1]. Systems of care for acute myocardial infarction had to be redesigned [2–5], ST-elevation myocardial infarction (STEMI) metrics and outcomes (including mortality) worsened [6–10], complications of late-presentation acute myocardial infarction increased [11], and lower hospitalization rates for acute coronary syndromes (ACS) [9, 12] paralleled an increase in the rates of out-of-hospital cardiac arrest (particularly among patients infected with COVID-19) [13].

While preliminary reports from small cohorts focused on STEMI [14–16] or ACS [10, 17] care at selected sites, systematic reporting from national or statewide registries of all-comers undergoing percutaneous coronary intervention (PCI) is scant [18]. Moreover, such reports only focused on the early phases of the pandemic ("first wave") and might have missed temporal changes occurring during later phases of this public health crisis. As such, a rigorous assessment of the impact of the COVID-19 pandemic on the delivery and outcomes of PCI in the general population is warranted.

The aim of the present study was to evaluate the temporal trends and outcomes of patients undergoing PCI in Michigan during the COVID-19 pandemic and to compare them with those of the pre-pandemic era, by using data from the Blue Cross Blue Shield of Michigan Cardiovascular Consortium (BMC2) PCI registry.

## Methods

### Study population

All patient data points were derived from a HIPAA-compliant database. The University of Michigan IRB has waived the need for ongoing IRB approval on all analysis that are performed using BMC2 data. Consent was not obtained as all data were analyzed anonymously. The study population consisted of consecutive patients who underwent PCI at all 48 non-federal hospitals in Michigan participating in the BMC2 registry between March 1, 2020 and December 31, 2020 (pandemic cohort). This population was compared with all patients who underwent PCI between March 1, 2019 and December 31, 2019 (pre-pandemic cohort). The inception date of the pandemic cohort was chosen based on the date when the first COVID-19 case was diagnosed in Michigan (March 11, 2020). Details of the BMC2 registry have been previously described [19–21]. Data, collected by on-site registered nurse coordinators, included

demographic and clinical characteristics, procedural details, and in-hospital outcomes of patients undergoing PCI procedures. Data quality and the inclusion of consecutive procedures were ensured by ad hoc queries, random chart reviews, detailed site audits by an experienced nurse auditor, and a series of diagnostic routines included in the database [22]. The registry was approved or the need for approval waived by the Institutional Review Board of each participating hospital. All relevant data necessary to replicate the study findings are within the paper. The authors are unable to share the raw data, due to contractual agreements between participating institutions and the BMC2 registry that prohibit data sharing with external agencies. However, the analysis code and metadata to support the study figures is available on request.

## Data definitions and clinical endpoints

The estimated glomerular filtration rate (eGFR) was calculated with the Chronic Kidney Disease Epidemiology Collaboration (CKD-EPI) equation [23]. Acute heart failure symptoms were defined as difficulty breathing, leg or feet swelling, pulmonary edema on chest X-ray or jugular venous distension. Cardiovascular instability was defined as a combination of cardiogenic shock, hemodynamic instability, persistent ischemic symptoms, acute heart failure symptoms, ventricular arrhythmia, and refractory shock. Cardiogenic shock was defined as a sustained (>30 min) episode of systolic blood pressure <90 mm Hg and/or cardiac index <2.2 l/min/m$^2$ determined to be secondary to cardiac dysfunction, and/or the requirement for intravenous inotropic or vasopressor agents or mechanical support to maintain blood pressure and cardiac index above those specified levels. These definitions were based on the NCDR CathPCI Registry v. 5.0 data dictionary.

The primary endpoint of this study was in-hospital death. Secondary endpoints included: acute kidney injury (AKI, defined as a ≥0.5 mg/dl absolute increase in serum creatinine from baseline [24, 25]); transfusion (at any time point between PCI and discharge); and major bleeding (defined as bleeding associated with a hemoglobin drop ≥5.0 g/dl from baseline).

## Statistical analysis

Continuous variables are reported as mean ± standard deviation (or as median and interquartile range, as appropriate), and categorical variables as number and percentage. The independent-samples Student's t-test was used to compare continuous variables with normal distribution, and the Wilcoxon signed rank test was used to compare continuous variables with a non-normal distribution. Chi-square test was used to compare differences between categorical variables. Standardized mean differences (SMDs) for each variable are also reported, since our large sample size might have led to statistically significant but clinically unimportant differences as assessed by the aforementioned tests. Traditionally, SMDs <10% have been considered to indicate a negligible difference in the mean or prevalence of a covariate between treatment groups, and thus would be unlikely to confound an analysis of clinical endpoints [26].

Temporal trends in PCI volumes (primary PCI for STEMI vs. all other PCI indications) were compared between the pre-pandemic and pandemic periods. A subgroup analysis was conducted, within the pandemic cohort, between patients with a positive vs. negative COVID-19 test.

Risk-adjusted comparisons of in-hospital outcomes (death, AKI, transfusion, and major bleeding) were performed between the pre-pandemic and pandemic periods, using logistic regression models adjusting for baseline patient predicted risk, estimated from a recently updated version of our random forest model [27]. In addition, we performed a sensitivity

analysis using patient risk estimates based on the recently published NCDR CathPCI Registry mortality risk model [28]. The results of these analyses are presented as odds ratios (OR) and 95% confidence intervals (CI).

A 2-tailed p-value <0.05 was considered statistically significant. All analyses were performed with R version 3.6.3 (R Core Team, Vienna, Austria).

## Results

### Overall patient population

During the pre-pandemic period, a total of 25,737 patients underwent PCI at facilities participating in the BMC2 Registry, which decreased to 21,822 subjects in the pandemic period, a 15.2% relative decrease. Fig 1 demonstrates the decrease in PCI volumes across all indications between the pre-pandemic and pandemic periods (-11.7% for stable coronary artery disease [CAD], -19.7% for non-ST-elevation ACS [NSTE-ACS], -12.6% for STEMI). Table 1 shows the baseline and procedural characteristics of the two cohorts. There were minimal differences between groups for the studied variables, with SMDs <10% in all cases. Notably, presentation with non-ST-elevation ACS, STEMI, and cardiovascular instability remained stable at ~41%, ~16%, and ~23%, respectively. Similarly, there were no differences in the prevalence of multi-vessel disease, left main PCI, type C lesions and need for mechanical circulatory support.

Table 2 shows the unadjusted rates of in-hospital outcomes of the two cohorts. There were no significant differences for most outcomes. However, the in-hospital death rate was marginally higher in the pandemic period (2.0% vs. 1.7%; SMD 2.5%), while the incidence of heart

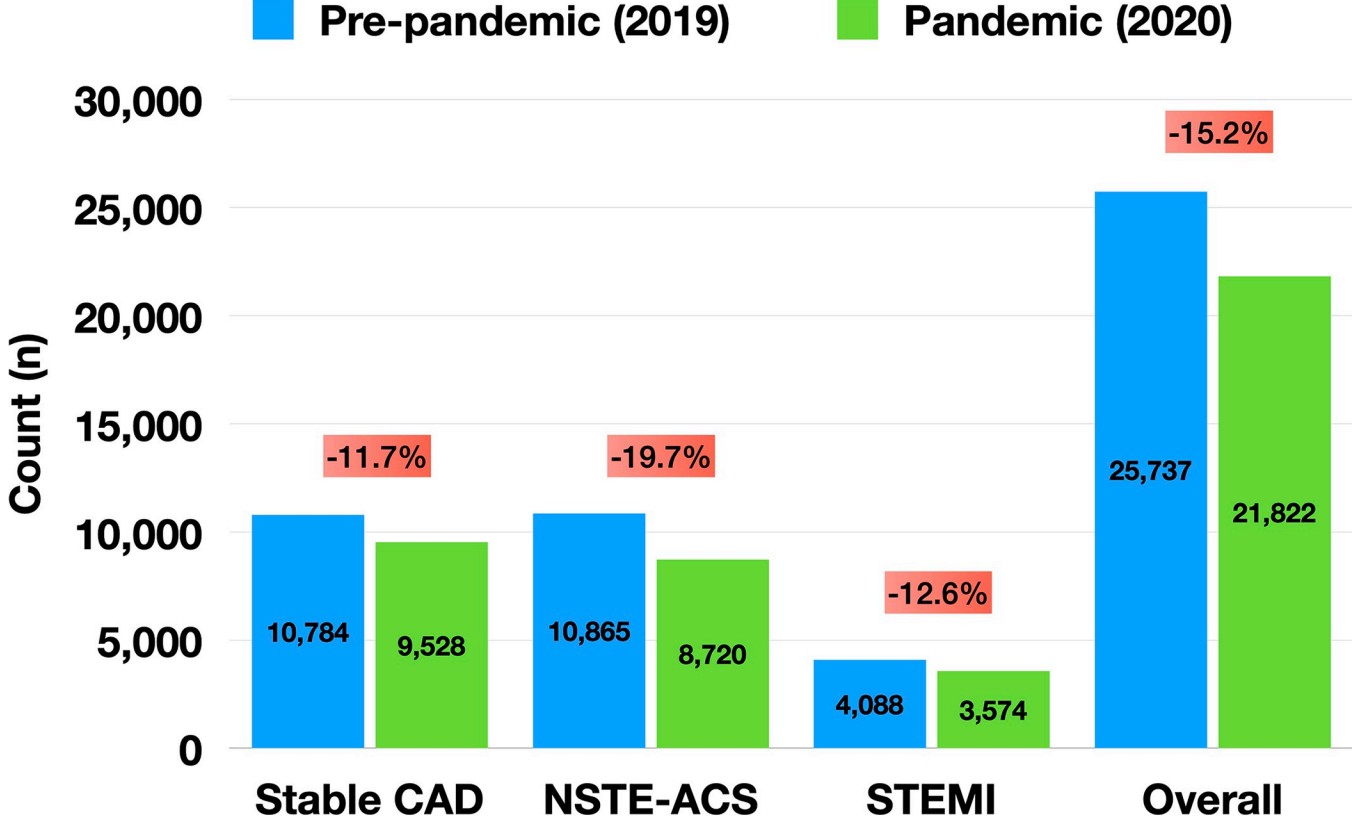

**Fig 1. Percutaneous coronary intervention (PCI) volumes for various indications in the pre-pandemic vs. pandemic period.** Abbreviations: CAD, coronary artery disease; NSTE-ACS, non-ST-elevation acute coronary syndrome; STEMI, ST-elevation myocardial infarction.

**Table 1. Clinical and procedural characteristics in the overall population.**

| | Overall (n = 47,559) | March-December 2019 Pre-pandemic (n = 25,737) | March-December 2020 Pandemic (n = 21,822) | p | SMD (%) |
|---|---|---|---|---|---|
| Age (years) | 66.6±11.7 | 66.7±11.8 | 66.5±11.7 | 0.135 | 1.4 |
| Body mass index (kg/m$^2$) | 30.7±7.2 | 30.7±7.4 | 30.7±7.1 | 1 | <0.001 |
| Sex (male) | 32,239 (67.8%) | 17,417 (67.7%) | 14,822 (67.9%) | 0.569 | 0.5 |
| Race (black) | 4,829 (10.2%) | 2,570 (10.0%) | 2,259 (10.4%) | 0.193 | 1.2 |
| Diabetes mellitus | 19,556 (41.1%) | 10,587 (41.1%) | 8,969 (41.1%) | 0.009 | <0.001 |
| Hypertension | 41,098 (86.4%) | 22,229 (86.4%) | 18,869 (86.5%) | 0.939 | 0.3 |
| Dyslipidemia | 39,218 (82.5%) | 21,007 (81.6%) | 18,211 (83.5%) | <0.001 | 4.9 |
| Current smoker | 11,135 (23.4%) | 6,036 (23.5%) | 5,099 (23.4%) | 0.833 | 0.2 |
| Prior myocardial infarction | 15,359 (32.3%) | 8,329 (32.4%) | 7,030 (32.2%) | 0.750 | 0.3 |
| Prior PCI | 21,378 (45.0%) | 11,578 (45.0%) | 9,800 (44.9%) | 0.851 | 0.2 |
| Prior CABG | 7,605 (16.0%) | 4,157 (16.2%) | 3,448 (15.8%) | 0.303 | 1.0 |
| Peripheral arterial disease | 6,627 (13.9%) | 3,623 (14.1%) | 3,004 (13.8%) | 0.338 | 0.9 |
| Cerebrovascular disease | 7,739 (16.3%) | 4,155 (16.1%) | 3,584 (16.4%) | 0.414 | 0.8 |
| Chronic lung disease | 9,260 (19.5%) | 5,011 (19.5%) | 4,249 (19.5%) | 0.717 | <0.001 |
| Chronic heart failure | 14,737 (31.0%) | 7,978 (31.0%) | 6,759 (31.0%) | 0.947 | 0.1 |
| Dialysis | 1,472 (3.1%) | 781 (3.0%) | 691 (3.2%) | 0.421 | 0.8 |
| Atrial fibrillation | 7,890 (16.6%) | 4,268 (16.6%) | 3,622 (16.6%) | 0.975 | <0.001 |
| eGFR (ml/min/1.73 m$^2$) | 70.7±24.0 | 70.9±23.9 | 70.6±24.0 | 0.203 | 1.2 |
| Hemoglobin (g/dl) | 13.4±2.0 | 13.4±2.0 | 13.4±2.1 | <0.001 | 3.6 |
| LVEF (%) | 51.2±14.0 | 51.3±14.0 | 51.1±14.1 | 0.131 | 1.8 |
| LVEF ≤35% | 4,979 (10.5%) | 2,640 (10.3%) | 2,339 (10.7%) | 0.105 | 1.5 |
| Presentation with NSTE-ACS | 19,585 (41.2%) | 10,865 (42.2%) | 8,720 (40.0%) | <0.001 | 4.6 |
| Presentation with STEMI | 7,662 (16.1%) | 4,088 (15.9%) | 3,574 (16.4%) | 0.148 | 1.3 |
| Cardiovascular instability | 10,752 (22.6%) | 5,840 (22.7%) | 4,912 (22.5%) | 0.645 | 0.4 |
| Cardiac arrest | 581 (1.2%) | 337 (1.3%) | 244 (1.1%) | 0.007 | 1.8 |
| Radial access | 26,280 (55.3%) | 13,913 (54.1%) | 12,367 (56.7%) | <0.001 | 5.3 |
| Multivessel disease | 21,578 (45.4%) | 11,682 (45.4%) | 9,896 (45.4%) | 0.943 | 0.1 |
| Left main PCI | 1,079 (2.7%) | 590 (2.7%) | 489 (2.7%) | 0.678 | 0.4 |
| Mechanical circulatory support | 1,731 (3.6%) | 930 (3.6%) | 801 (3.7%) | 0.759 | 0.3 |
| Type C lesion | 32,100 (67.5%) | 17,164 (66.7%) | 14,936 (68.4%) | <0.001 | 3.7 |
| Dose-area product (Gy·cm$^2$) | 104±176 | 107±167 | 101±185 | <0.001 | 3.5 |
| Air Kerma (Gy) | 1.4±1.4 | 1.5±1.4 | 1.4±1.4 | <0.001 | 7.2 |
| Contrast volume (ml) | 141±60 | 144±61 | 139±60 | <0.001 | 8.5 |

Values are expressed as mean ± standard deviation, or n (%). Abbreviations: CABG, coronary artery bypass graft; eGFR, estimated glomerular filtration rate; LVEF, left ventricular ejection fraction; NSTE-ACS, non-ST-elevation acute coronary syndrome; PCI, percutaneous intervention; SMD, standardized mean difference; STEMI, ST-elevation myocardial infarction.

failure (2.2% vs. 1.8%; SMD 2.6%) and myocardial infarction (0.6% vs. 0.4%; SMD 2.1%) was marginally higher in the pre-pandemic cohort.

## Temporal trends in PCI for STEMI vs. all other indications and primary PCI mortality

Fig 2 shows monthly volumes for primary PCI and PCI for all other indications during 2019 and 2020, in relation to COVID-19 case count in the State of Michigan (which was

**Table 2. In-hospital outcomes in the overall population.**

| | Overall (n = 47,559) | March-December 2019 Pre-pandemic (n = 25,737) | March-December 2020 Pandemic (n = 21,822) | p | SMD (%) |
|---|---|---|---|---|---|
| Heart failure | 938 (2.0%) | 549 (2.2%) | 389 (1.8%) | 0.005 | 2.6 |
| Myocardial infarction | 246 (0.5%) | 151 (0.6%) | 95 (0.4%) | 0.025 | 2.1 |
| Cardiogenic shock | 804 (1.7%) | 432 (1.7%) | 372 (1.7%) | 0.869 | 0.2 |
| Acute kidney injury | 1,177 (3.2%) | 623 (3.1%) | 554 (3.4%) | 0.056 | 2.0 |
| New requirement for dialysis | 195 (0.4%) | 97 (0.4%) | 98 (0.4%) | 0.251 | 1.1 |
| Major bleeding | 461 (1.0%) | 255 (1.0%) | 206 (0.9%) | 0.637 | 0.5 |
| Transfusion | 1,238 (2.6%) | 671 (2.6%) | 567 (2.6%) | 0.975 | 0.1 |
| Tamponade | 71 (0.1%) | 37 (0.1%) | 34 (0.2%) | 0.830 | 0.3 |
| Stroke | 209 (0.4%) | 104 (0.4%) | 105 (0.5%) | 0.231 | 1.2 |
| Death | 879 (1.8%) | 435 (1.7%) | 444 (2.0%) | 0.006 | 2.5 |

Values are expressed as n (%)

downloaded from: https://data.cdc.gov/Case-Surveillance/United-States-COVID-19-Cases-and-Deaths-by-State-o/9mfq-cb36). Overall, primary PCI volumes were lower for almost any given month in 2020, compared with 2019. There was a sharp decline in primary PCI between February and May 2020, which began slightly before the first diagnosed COVID-19 case in Michigan on March 11, 2020. Subsequent months saw a partial recovery in procedural volumes. A similar, although much more pronounced, pattern was observed with regard to PCI volume for all other indications. In particular, in April 2020 PCI volume was less than half of what had been in April 2019.

Fig 3 displays monthly mortality rates for primary PCI in the pre-pandemic and pandemic periods. Except for June and July, mortality rates were higher in 2020 compared with 2019. As shown in S1 Fig, there were clinically negligible differences in the door-to-balloon times for both cases with (pre-pandemic 113 (94–145) vs. pandemic 115.5 (95–146) min, p = 0.582) and without (pre-pandemic 68 (53–84) vs. pandemic 69 (55–86) min, p = 0.002) transfer from another facility. The overall symptoms-to-balloon time slightly increased from the pre-pandemic to the pandemic period: 158 (121–228.5) min vs. 166 (123–245.5) min (p = 0.007).

## Subgroup analysis of COVID-19 positive vs. negative patients in the pandemic cohort

S2 Fig presents COVID-19 status of the patients in the pandemic cohort. The chart highlights how COVID-19 testing slowly became more widespread throughout 2020, so that by the end of the year approximately two-thirds of patients undergoing PCI underwent testing. Table 3 shows a comparison of the clinical and procedural characteristics of patients who underwent PCI in the pandemic period and had a positive (n = 93) vs. negative (n = 9,935) COVID-19 test. There were important differences between the two groups. COVID-19 patients had a higher prevalence of black race (24.7% vs. 11.5%, SMD 34.9%), diabetes mellitus (51.6% vs. 41.9%, SMD 19.5%), presentation with STEMI (35.5% vs. 12.8%, SMD 55.1%), cardiac arrest (4.3% vs. 0.9%, SMD 21.9%), multivessel disease (55.9% vs. 46.5%, SMD 19.0%), and need for mechanical circulatory support (8.6% vs. 3.9%, SMD 19.5%).

Table 4 shows the in-hospital outcomes according to COVID-19 status. There were important differences between groups for most outcomes. In particular, patients with COVID-19 suffered higher rates of death (12.9% vs. 2.0%, p<0.001, SMD 42.3%), AKI (8.8% vs. 3.9%,

## Primary PCI procedures

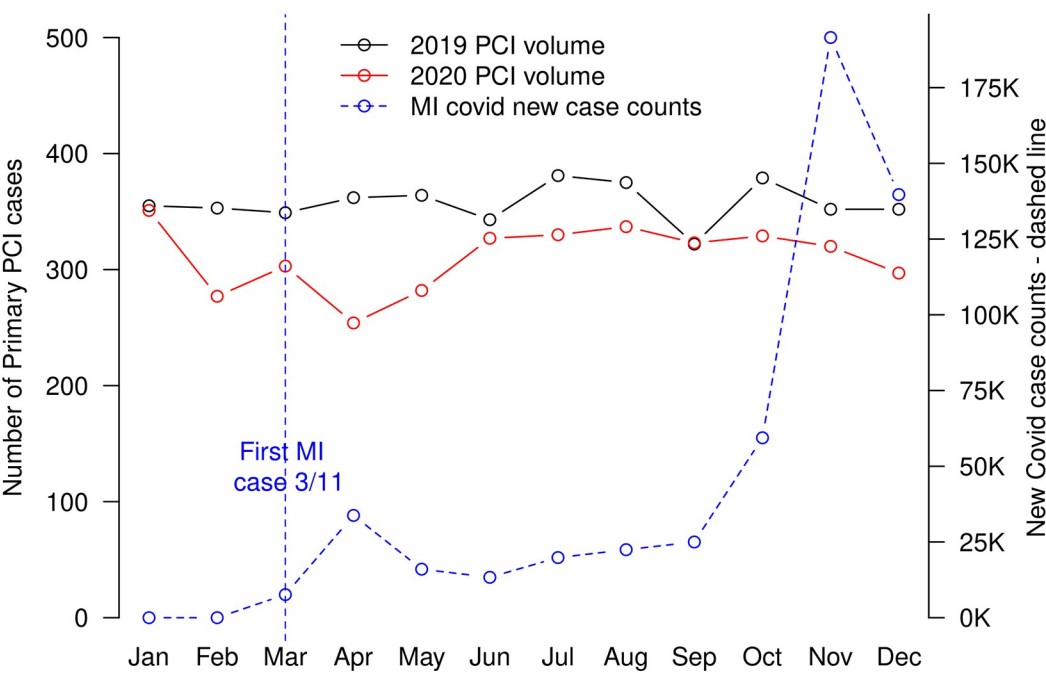

## All other PCI procedures

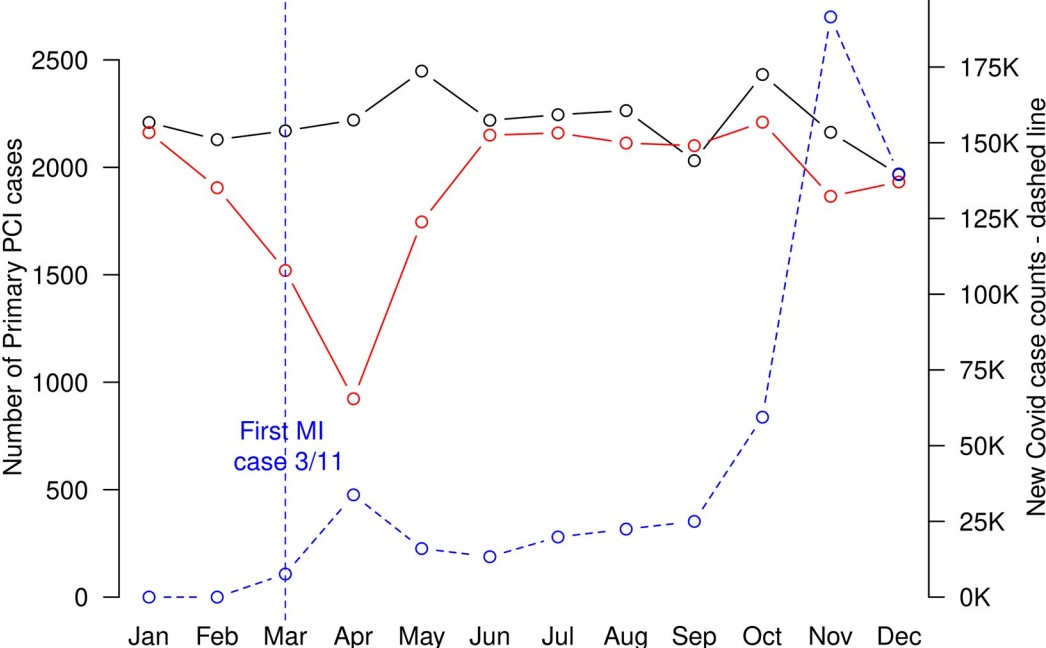

**Fig 2. Monthly primary and non-primary PCI volume in the pre-pandemic (black) vs. pandemic (red) period.** Monthly COVID-19 case counts (blue) are also displayed. The first COVID-19 case in Michigan was diagnosed on March 11, 2020.

## Primary PCI mortality by month

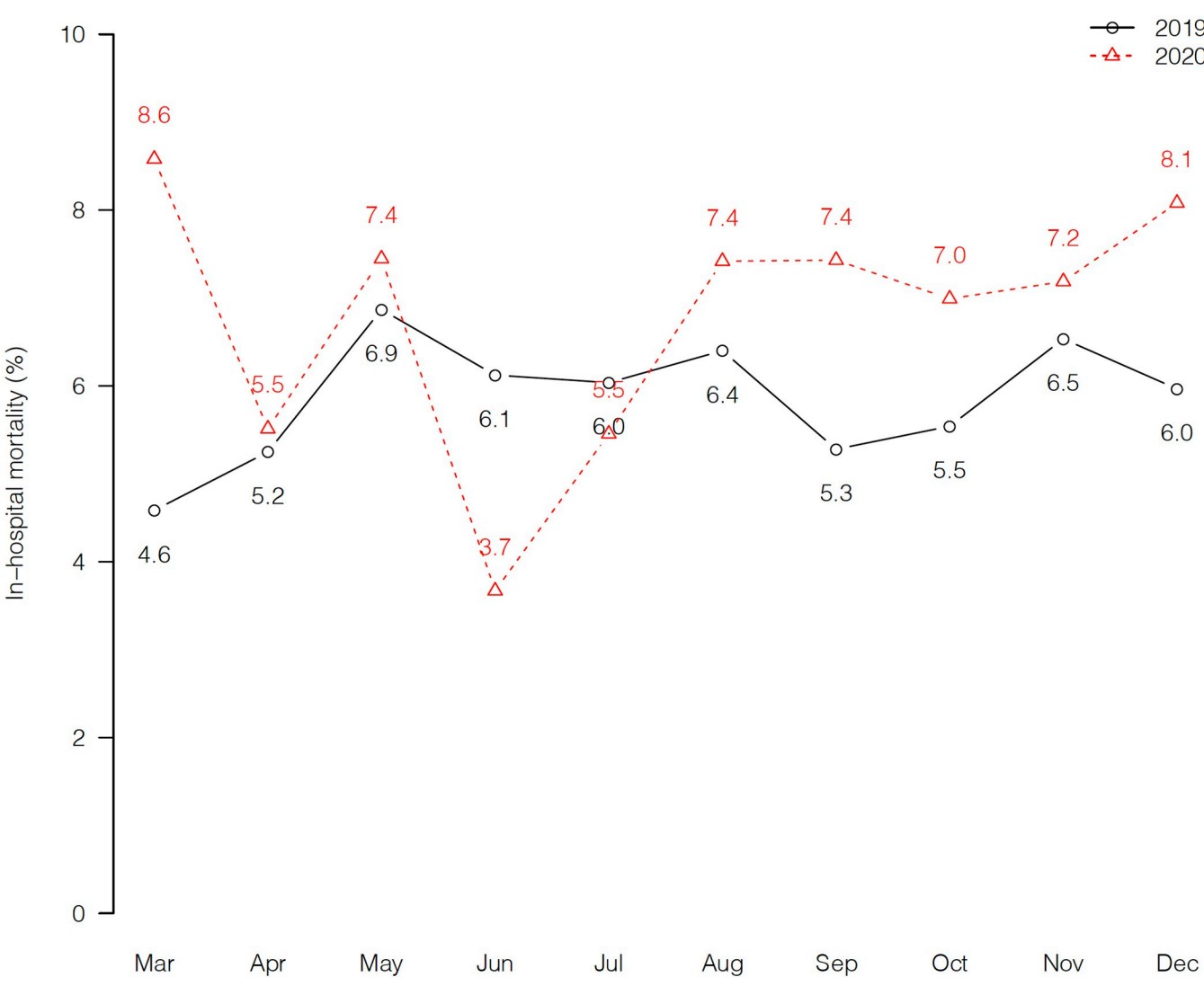

**Fig 3. Primary PCI in-hospital mortality by month in the pre-pandemic (black) vs. pandemic (red) period.**

p = 0.050, SMD 20.3%), new requirement for dialysis (3.2% vs. 0.4%, p = 0.002, SMD 20.8%), and transfusion (8.6% vs. 2.9%, p = 0.003, SMD 24.8%).

### Adjusted comparisons of in-hospital outcomes

**Fig 4** presents the risk-adjusted in-hospital outcome comparisons between the pre-pandemic and pandemic period, as well as between subjects with a positive vs. negative COVID-19 test in the 2020 cohort. Patients in the pandemic cohort suffered a higher adjusted risk of all-cause death (BMC2 model: OR 1.26, 95% CI 1.07–1.47, p = 0.005; new NCDR model: OR 1.22, 95% CI 1.05–1.41, p = 0.011), while differences in AKI, transfusion and major bleeding were non-significant. On the other hand, COVID-19 patients in 2020 had higher adjusted risk of all-cause death (BMC2 model: OR 5.69, 95% CI 2.54–12.74, p<0.001; new NCDR model: OR 5.28, 95% CI 2.42–11.53, p<0.001) and transfusion (OR 3.32, 95% CI 1.52–7.26, p = 0.003),

**Table 3. Clinical and procedural characteristics in patients with a positive vs. negative COVID-19 test in the pandemic period.**

| | Overall (n = 10,028) | COVID-19 negative (n = 9,935) | COVID-19 positive (n = 93) | *p* | SMD (%) |
|---|---|---|---|---|---|
| Age (years) | 66.9±11.7 | 67.0±11.7 | 66.0±13.1 | 0.446 | 7.5 |
| Body mass index (kg/m$^2$) | 30.6±6.9 | 30.6±6.9 | 30.9±6.5 | 0.719 | 3.9 |
| Sex (male) | 6,681 (66.6%) | 6,615 (66.6%) | 66 (71.0%) | 0.434 | 9.5 |
| Race (black) | 1,164 (11.6%) | 1,141 (11.5%) | 23 (24.7%) | <0.001 | 34.9 |
| Diabetes mellitus | 4,212 (42.0%) | 4,164 (41.9%) | 48 (51.6%) | 0.075 | 19.5 |
| Hypertension | 8,758 (87.3%) | 8,681 (87.4%) | 77 (82.8%) | 0.242 | 12.9 |
| Dyslipidemia | 8,408 (83.9%) | 8,335 (83.9%) | 73 (78.5%) | 0.203 | 13.9 |
| Current smoker | 2,269 (22.6) | 2,260 (22.7%) | 9 (9.7%) | 0.004 | 36.0 |
| Prior myocardial infarction | 3,255 (32.5%) | 3,232 (32.6%) | 23 (25.0%) | 0.153 | 16.7 |
| Prior PCI | 4,530 (45.2%) | 4,497 (45.3%) | 33 (35.5%) | 0.073 | 20.1 |
| Prior CABG | 1,625 (16.2%) | 1,614 (16.3%) | 11 (11.8%) | 0.311 | 12.8 |
| Peripheral arterial disease | 1,474 (14.7%) | 1,467 (14.8%) | 7 (7.5%) | 0.069 | 23.2 |
| Cerebrovascular disease | 1,713 (17.1%) | 1,703 (17.2%) | 10 (10.8%) | 0.135 | 18.6 |
| Chronic lung disease | 2,067 (20.6%) | 2,053 (20.7%) | 14 (15.1%) | 0.228 | 14.7 |
| Chronic heart failure | 3,440 (34.3%) | 3,415 (34.4%) | 25 (26.9%) | 0.159 | 16.3 |
| Dialysis | 358 (3.6%) | 355 (3.6%) | 3 (3.2%) | 1 | 1.9 |
| Atrial fibrillation | 1,700 (17.0%) | 1,688 (17.0%) | 12 (12.9%) | 0.365 | 11.5 |
| eGFR (ml/min/1.73 m$^2$) | 70.4±24.7 | 70.4±24.7 | 67.9±28.6 | 0.329 | 9.5 |
| Hemoglobin (g/dl) | 13.3±2.1 | 13.3±2.1 | 13.1±2.1 | 0.284 | 11.3 |
| LVEF (%) | 50.8±14.4 | 50.8±14.4 | 50.4±12.4 | 0.816 | 3.5 |
| LVEF ≤35% | 1,203 (12.0%) | 1,196 (12.0%) | 7 (7.5%) | 0.241 | 15.2 |
| Presentation with NSTE-ACS | 4,520 (45.1%) | 4,472 (45.0%) | 48 (51.6%) | 0.243 | 13.2 |
| Presentation with STEMI | 1,301 (13.0%) | 1,268 (12.8%) | 33 (35.5%) | <0.001 | 55.1 |
| Cardiovascular instability* | 1,973 (19.7%) | 1,933 (19.5%) | 40 (43.0%) | <0.001 | 52.5 |
| Cardiac arrest | 89 (0.9%) | 85 (0.9%) | 4 (4.3%) | 0.003 | 21.9 |
| Radial access | 5,470 (54.6%) | 5,434 (54.7%) | 36 (38.7%) | 0.003 | 32.5 |
| Multivessel disease | 4,666 (46.5%) | 4,614 (46.5%) | 52 (55.9%) | 0.086 | 19.0 |
| Left main PCI | 277 (3.3%) | 275 (3.3%) | 2 (2.4%) | 0.900 | 5.2 |
| Mechanical circulatory support | 395 (3.9%) | 387 (3.9%) | 8 (8.6%) | 0.040 | 19.5 |
| Type C lesion | 6,954 (69.3%) | 6,888 (69.3%) | 66 (71.0%) | 0.820 | 3.6 |
| Dose-area product (Gy·cm$^2$) | 99±159 | 99±159 | 104±96 | 0.776 | 3.8 |
| Air Kerma (Gy) | 1.4±1.4 | 1.4±1.4 | 1.5±1.4 | 0.378 | 9.4 |
| Contrast volume (ml) | 139±60 | 139±60 | 146±59 | 0.268 | 11.6 |

Values are expressed as mean ± standard deviation, or n (%).

* Cardiovascular instability includes: cardiogenic shock, hemodynamic instability, persistent ischemic symptoms, acute heart failure symptoms, ventricular arrhythmia, and refractory shock.

Abbreviations: CABG, coronary artery bypass graft; eGFR, estimated glomerular filtration rate; LVEF, left ventricular ejection fraction; NSTE-ACS, non-ST-elevation acute coronary syndrome; PCI, percutaneous intervention; SMD, standardized mean difference; STEMI, ST-elevation myocardial infarction.

compared with subjects with a negative COVID-19 test. To ascertain whether the higher adjusted risk of mortality in the pandemic cohort was driven by inferior outcomes in COVID-19 patients, we performed a sensitivity analysis excluding patients with a positive COVID-19 test in the pandemic cohort. As reported in S3 Fig, there remained an increase in the adjusted risk of death in the pandemic cohort (OR 1.23, 95% CI 1.05–1.44, p = 0.012), thus indicating that the observed higher mortality risk in the pandemic cohort was associated with worse outcomes in both COVID-19 positive and negative patients. S4 Fig presents a risk-adjusted

**Table 4. In-hospital outcomes in patients with a positive vs. negative COVID-19 test in the pandemic period.**

| | Overall (n = 10,028) | COVID-19 negative (n = 9,935) | COVID-19 positive (n = 93) | p | SMD (%) |
|---|---|---|---|---|---|
| Heart failure | 163 (1.6%) | 159 (1.6%) | 4 (4.3%) | 0.102 | 16.0 |
| Myocardial infarction | 46 (0.5%) | 46 (0.5%) | 0 | 1 | 9.6 |
| Cardiogenic shock | 143 (1.4%) | 141 (1.4%) | 2 (2.2%) | 0.879 | 5.5 |
| Acute kidney injury | 297 (3.9%) | 290 (3.9%) | 7 (8.8%) | 0.050 | 20.3 |
| New requirement for dialysis | 47 (0.5%) | 44 (0.4%) | 3 (3.2%) | 0.002 | 20.8 |
| Major bleeding | 87 (0.9%) | 85 (0.9%) | 2 (2.2%) | 0.436 | 10.7 |
| Transfusion | 294 (2.9%) | 286 (2.9%) | 8 (8.6%) | 0.003 | 24.8 |
| Tamponade | 16 (0.2%) | 16 (0.2%) | 0 | 1 | 5.7 |
| Stroke | 60 (0.6%) | 60 (0.6%) | 0 | 0.939 | 11.0 |
| Death | 214 (2.1%) | 202 (2.0%) | 12 (12.9%) | <0.001 | 42.3 |

Values are expressed as n (%)

mortality comparison between the pre-pandemic and pandemic periods stratified by clinical presentation. While there was no difference in the outcome of patients with non-ACS presentation in the pandemic cohort (OR 1.02, 95% CI 0.66–1.58, p = 0.917), we observed a borderline higher risk of mortality for NSTE-ACS presentation (OR 1.23, 95% CI 0.95–1.60, p = 0.118) and a significantly higher mortality risk for STEMI patients (OR 1.34, 95% CI 1.07–1.68, p = 0.010) in the pandemic cohort.

## Discussion

The main findings of our study are: 1) there was a ~15% decrease in overall PCI volume from the pre-pandemic to the pandemic period, which was observed for all presentations (stable CAD, NSTE-ACS, and STEMI) and was particularly marked in the winter and spring months; 2) despite similar clinical and procedural characteristics of the study population in the two periods, monthly mortality rates for primary PCI were in general higher in the pandemic period, a finding that was however not related to clinically significant system delays in STEMI care; 3) in the pandemic period, COVID-19 positive patients undergoing PCI had markedly higher prevalence of baseline clinical and presentation features indicative of higher severity of illness (including STEMI, cardiac arrest, multivessel disease and need for mechanical circulatory support), which was associated with worse clinical outcomes; 4) compared with patients undergoing PCI in the pre-pandemic period, those undergoing PCI in the pandemic period had a higher risk of death, a finding that was only partially explained by worse outcomes observed in COVID-19 patients, and seemed to be more pronounced in subjects presenting with ACS (particularly STEMI).

Compared with prior literature, our report presents several unique strengths: 1) it provides in-depth insights on the outcomes of all-comers undergoing PCI before and during the COVID-19 pandemic; 2) it gives a representative snapshot of the outcomes of patients undergoing PCI in the whole State of Michigan (population: 10 million), thus avoiding issues related to selection bias; 3) our analysis covers a longer period compared to initial reports that exclusively focused on the "first wave" of the pandemic, thus providing additional insights on the evolution of these complex phenomena over time.

Kwok et al. [18] reported a sharp decrease (49%) in the overall PCI volume in England after the March 23, 2020 lockdown. This was particularly pronounced for patients undergoing PCI for stable CAD. These observations parallel our study findings. However, the authors reported a lower risk profile in patients undergoing PCI after the lockdown (particularly for

**A**

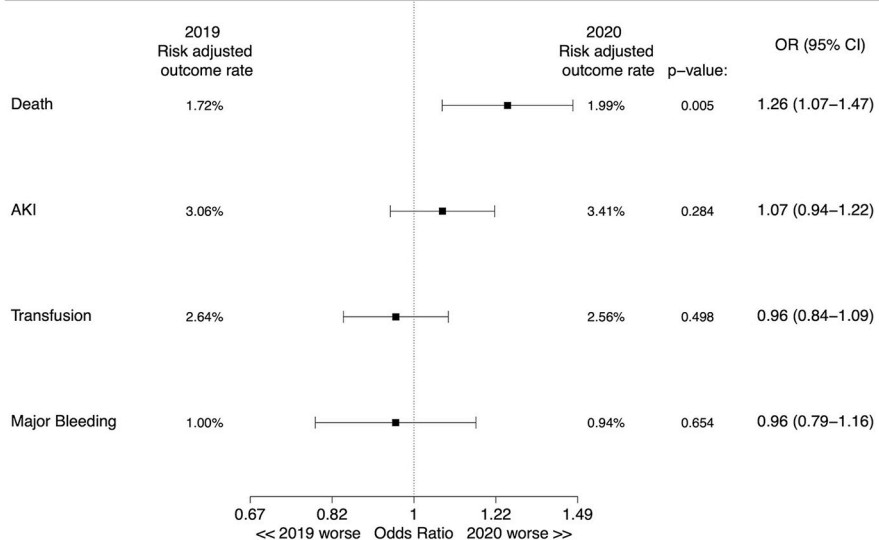

**B**

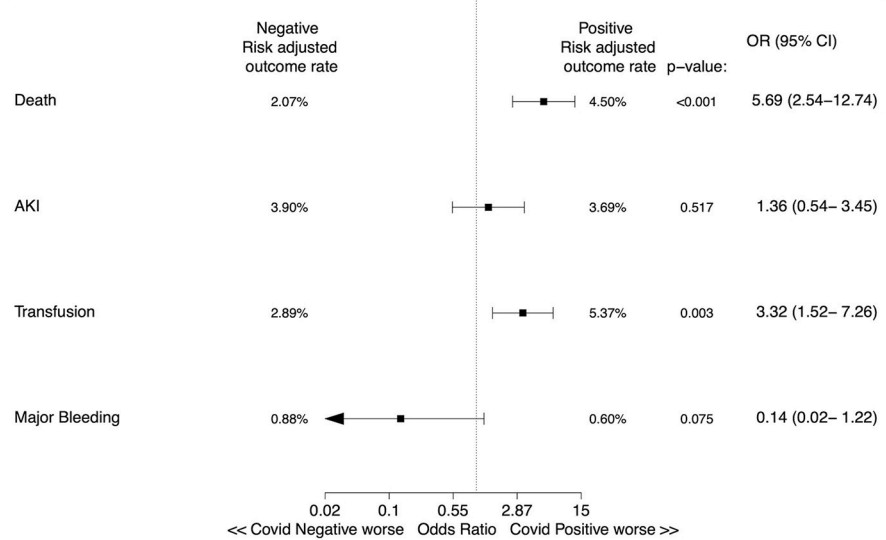

**Fig 4.** Risk-adjusted outcome comparison between (A) the pre-pandemic and pandemic period, and (B) within the pandemic cohort, between COVID-19 positive and negative patients. Abbreviations: AKI, acute kidney injury; CI, confidence interval; OR, odds ratio.

NSTE-ACS), thus reflecting a more conservative approach towards older patients and those with comorbidities. In contrast, we did not observe a change in clinical and procedural characteristics between the pre-pandemic and pandemic period. Further insights and comparisons are limited by the lack of COVID-19 status information and in-hospital outcomes, as well as the shorter inclusion period (until April 2020), in the study by Kwok et al. [18].

Garcia et al. [14] reported on the outcomes of COVID-19 patients presenting with STEMI at selected sites in North America. Similar to our findings, they observed a higher prevalence of minority ethnicities (Hispanics and Blacks), baseline comorbidities, as well as adverse clinical presentation (cardiogenic shock), in COVID-19 positive patients, which paralleled a markedly increased incidence of a composite endpoint of in-hospital death, stroke, recurrent myocardial infarction, or repeat unplanned revascularization in such group, compared with COVID-19 negative subjects (36% vs. 5%, p<0.001). These observations are to be put into the context of a 38% reduction in STEMI catheterization laboratory activations in the first quarter of 2020 in the U.S. [15]. A marked decrease in the rates of admission for STEMI and NSTE-ACS was also reported in Northern Italy during the first month of the pandemic [12], and COVID-19 patients presenting with STEMI from the same geographic area suffered a very high mortality (39%) [16]. Similarly, Kite et al. [10] found a higher risk of mortality in COVID-19 patients presenting with ACS in a propensity score-adjusted comparison with pre-pandemic ACS patients.

Mohamed et al. [29] reported a marked decrease in cardiac procedures in England between January and May 2020. Cardiac catheterization and device implantations were the most affected in terms of absolute numbers. Of note, for these two procedures an increase in 30-day mortality was observed, suggesting that perhaps those were performed in higher-risk patients for whom deferral (or non-invasive evaluation, in case of cardiac catheterization) was not possible.

We found that monthly mortality rates for patients undergoing PCI for STEMI were higher in the pandemic period (except for June and July 2020). Data from Hong Kong during the first two weeks of the pandemic highlight an important prolongation of STEMI time metrics such as symptom-to-first-medical-contact and door-to-balloon times [3]. However, such an issue was not observed in our cohort, where door-to-balloon times were prolonged by ~1–2 minutes only (**S1 Fig**) and overall symptoms-to-balloon time suffered a median increase of just 8 minutes. Therefore, differences in system delays for STEMI care were unlikely to be responsible for the differences in outcomes during the first 9 months of the pandemic in Michigan and possibly indicated effective organizational restructuring.

In an effort to identify the reason underlying the higher risk-adjusted mortality in all-comers undergoing PCI during the pandemic, we performed additional sensitivity and stratified analyses, which demonstrated that such finding was more pronounced in patients presenting with ACS (particularly STEMI) and was only partially explained by worse outcomes observed in subjects with COVID-19. Several authors have analyzed the "collateral damage" of the COVID-19 pandemic to cardiac care in non-COVID-19 subjects. Moroni et al. [11] first reported on the phenomenon of medical care avoidance among ACS patients during the first weeks of the pandemic in Italy. Early in the course of the pandemic, public health officials and the media were discouraging the population from seeking care in the emergency room setting to limit the spread of COVID-19. In this context, several patients who were not infected with COVID-19 suffered complications of myocardial infarction associated with late presentation (left ventricular thrombosis with systemic embolization, cardiogenic shock, papillary muscle and free-wall rupture, etc.). Further adjustments in public health information of the general population, as well as better COVID-19 containment measures, have likely mitigated this phenomenon. However, many have speculated that the pandemic might exert a persistent, longer-

term effect in reducing the access to state-of-the-art care for life-threatening cardiac conditions, such as aortic stenosis or complex, multivessel or left main CAD [29, 30].

## Limitations

This is a retrospective study, and it is susceptible to limitations ascribed to such a study design. In particular, no causality can be claimed for any of the clinical associations we observed. The subanalysis in the pandemic cohort according to COVID-19 status is limited by the fact that COVID-19 testing was performed in only 46% of patients in such cohort due to limited test availability, particularly in the earlier months of the pandemic. While we were able to rule out specific causes underlying our key study findings (e.g., that the higher risk-adjusted mortality observed in the pandemic cohort could be exclusively linked to worse outcomes in COVID-19 patients), we were not able to positively identify the reasons for such findings, and the theory of the "collateral damage" of the pandemic on cardiovascular care delivery in non-COVID-19 patients remains purely speculative. The data from Michigan reflect the experience of a long-standing quality improvement collaborative that actively shared best practices for catheterization laboratory response to COVID and may not be generalizable to all health systems. Finally, our registry included only patients who actually underwent PCI and we cannot provide insights on the volume trends and outcomes of patients who presented with ACS or stable CAD but who did not undergo PCI.

## Conclusions

We observed a ~15% reduction in PCI volumes for all indications between March and December 2020 in Michigan. Although patient and procedural characteristics remained essentially stable between 2019 and 2020, higher risk-adjusted mortality was observed during the pandemic, a finding that was not completely explained by worse outcomes in COVID-19 patients and was more pronounced in subjects presenting with ACS (particularly STEMI), raising the possibility of an indirect effect of the pandemic on cardiovascular care delivery in non-COVID-19 patients. In the pandemic cohort, COVID-19 patients suffered higher risk-adjusted mortality.

## Supporting information

**S1 Fig. Boxplots comparing door-to-balloon times for primary PCI in the pre-pandemic (2019) vs. pandemic (2020) period.**
(TIF)

**S2 Fig. COVID-19 status of the patients in the pandemic (2020) cohort.**
(TIF)

**S3 Fig. Risk-adjusted in-hospital outcomes in the pre-pandemic (2019) vs. pandemic (2020) periods, excluding COVID-19-positive patients in the 2020 cohort.**
(TIF)

**S4 Fig. Risk-adjusted mortality comparison between the pre-pandemic (2019) and pandemic (2020) periods stratified by clinical presentation.**
(TIF)

## Author Contributions

**Conceptualization:** Lorenzo Azzalini, Devraj Sukul, Hitinder S. Gurm.

**Data curation:** Milan Seth.

**Formal analysis:** Milan Seth.

**Funding acquisition:** Hitinder S. Gurm.

**Investigation:** Lorenzo Azzalini, Devraj Sukul, Javier A. Valle, Edouard Daher, Brett Wanamaker, Michael T. Tucciarone, Anwar Zaitoun, Ryan D. Madder, Hitinder S. Gurm.

**Methodology:** Milan Seth, Hitinder S. Gurm.

**Project administration:** Milan Seth, Hitinder S. Gurm.

**Supervision:** Lorenzo Azzalini, Hitinder S. Gurm.

**Writing – original draft:** Lorenzo Azzalini, Milan Seth.

**Writing – review & editing:** Devraj Sukul, Javier A. Valle, Edouard Daher, Brett Wanamaker, Michael T. Tucciarone, Anwar Zaitoun, Ryan D. Madder, Hitinder S. Gurm.

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
