## [Decision Letter · Decision Letter 0]

29 Mar 2022

PONE-D-21-35320

Trends and outcomes of percutaneous coronary intervention during the COVID-19 pandemic in Michigan

PLOS ONE

Dear Dr. Gurm,

Thank you for submitting your manuscript to PLOS ONE. After careful consideration, we feel that it has merit but does not fully meet PLOS ONE’s publication criteria as it currently stands. Therefore, we invite you to submit a revised version of the manuscript that addresses the points raised during the review process.

ACADEMIC EDITOR: Thank you very much for having submitted this paper for consideration. The reviewers raised some comments and in particular reviewer number 2. The following indications will help you to improve the quality of your paper.

We look forward to receiving your revised manuscript.

Kind regards,

Simone Savastano

Academic Editor

PLOS ONE

“This work was supported by the Blue Cross Blue Shield of Michigan and Blue Care Network as part of the Blue Cross Blue Shield of Michigan Value Partnerships program. The funding source supported data collection at each site and funded the data-coordinating center, but had no role in study concept, interpretation of findings, or in the preparation, final approval or decision to submit the manuscript.”

“This work was supported by the Blue Cross Blue Shield of Michigan and Blue Care Network as part of the Blue Cross Blue Shield of Michigan Value Partnerships program. The funding source supported data collection at each site and funded the data-coordinating center, but had no role in study concept, interpretation of findings, or in the preparation, final approval or decision to submit the manuscript.”

“I have read the journal's policy and the authors of this manuscript have the following competing interests: Dr. Azzalini received consulting fees from Teleflex, Abiomed, Asahi Intecc, Abbott Vascular, Philips, and Cardiovascular Systems, Inc. Dr. Sukul receives salary support from the Blue Cross Blue Shield of Michigan for his role in BMC2. Dr. Gurm receives research support from Blue Cross and Blue Shield of Michigan, and Michigan Translational Research and Commercialization for Life Sciences Innovation Hub. He is the co-founder of, owns equity in, and is a consultant to Amplitude Vascular Systems. He also owns equity in Jiaxing Bossh Medical Technology Partnership and is a consultant for Osprey Medical. He is the chair of the Clinical Events Committee for the PERFORMANCE trial sponsored by Contego Medical. The other authors have no disclosures.”

Additional Editor Comments:

Thank you very much for having submitted this paper for consideration. The reviewers raised some comments and in particular reviewer number 2. The following indications will help you to improve the quality of your paper.

Reviewers' comments:

Reviewer's Responses to Questions

**Comments to the Author**

1. Is the manuscript technically sound, and do the data support the conclusions?

Reviewer #1: Yes

Reviewer #2: Yes

2. Has the statistical analysis been performed appropriately and rigorously? 

Reviewer #1: Yes

Reviewer #2: Yes

3. Have the authors made all data underlying the findings in their manuscript fully available?

Reviewer #1: Yes

Reviewer #2: Yes

4. Is the manuscript presented in an intelligible fashion and written in standard English?

Reviewer #1: Yes

Reviewer #2: Yes

5. Review Comments to the Author

Reviewer #1: Azzalini et al. report an interesting study regarding the evaluation of the influence of the COVID-19 pandemic on temporal trends and outcomes of patients undergoing percutaneous coronary intervention (PCI) during the year 2020 compared to year before (2019). The topic is of high interest due the persistence of the COVID pandemic that is bringing the world health to its knees. Despite numerous previous manuscripts depicting differences in PCI volumes and outcomes between the pandemic and the pre-pandemic era this paper covers a longer period compared to initial reports that exclusively focused on the “first wave” of the pandemic. Furthermore, it gives in-depth insights on the outcomes of all-comers undergoing PCI before and during the COVID-19 pandemic, showing that the worse in-hospital outcomes are not exclusively related to the COVID infection itself but also probably to the “collateral damage” of the COVID-19 pandemic to the entire cardiac care system.

The manuscript is well written, very interesting with a robust statistical methodology. Therefore, I do not have further revisions to propose.

Reviewer #2: The aim of this retrospective, multicenter study was to assess outcomes of patients undergoing percutaneous coronary intervention (PCI) in Michigan during the COVID-19 pandemic in 2020 (between March and December), comparing them to outcomes in the same period of the previous year. The Authors used data from the Blue Cross Blue Shield of Michigan Cardiovascular Consortium (BMC2) PCI registry which included data from all the catheterization laboratories in Michigan. The Authors found a ~15% decrease in overall PCI volume from the pre-pandemic to the pandemic period; monthly mortality rates for primary PCI were in general higher in the pandemic period. Furthermore, patients undergoing PCI in the pre-pandemic period, those undergoing PCI in the pandemic period had a higher risk of death. This last finding was more pronounced in patients presenting with ST-segment elevation myocardial infarction (STEMI).

The topic is interesting, the analyses are elegant and the manuscript clearly written.

I have some considerations:

• Monthly mortality rates for primary PCI were in general higher in the pandemic period, a finding that was however not related to clinically significant system delays in STEMI care. As already mentioned by the Authors, one of the possible interpretations is that patients tried to avoid contacting the emergency services for fear of entering the hospitals. Indeed, it is possible that predominantly patients with very severe symptomatology contacted the emergency services or went to the hospital. In this context, it is also possible that patients waited until they had an advanced clinical situation before asking for assistance. It is known that several patients who were not infected with COVID-19 suffered complications of myocardial infarction associated with late presentation. It would be interesting to add “pain-to-balloon” to the data shown.

• It is important to note also the low absolute number of STEMIs and NSTEMIs in 2020 compared to 2019: perhaps another indication that many patients stayed home and only the most symptomatic called for rescue.

• The authors performed outcome analyses using in-hospital mortality. Although the latter is widely used in the literature, I believe it would be more elegant and objective to use 30-day mortality, if it were available.

6. PLOS authors have the option to publish the peer review history of their article (what does this mean?). If published, this will include your full peer review and any attached files.

Reviewer #1: **Yes: **Gianmarco Iannopollo

Reviewer #2: No

---

## [Author Response · Author response to Decision Letter 0]

29 Jun 2022

Reviewer #1: Azzalini et al. report an interesting study regarding the evaluation of the influence of the COVID-19 pandemic on temporal trends and outcomes of patients undergoing percutaneous coronary intervention (PCI) during the year 2020 compared to year before (2019). The topic is of high interest due the persistence of the COVID pandemic that is bringing the world health to its knees. Despite numerous previous manuscripts depicting differences in PCI volumes and outcomes between the pandemic and the pre-pandemic era this paper covers a longer period compared to initial reports that exclusively focused on the “first wave” of the pandemic. Furthermore, it gives in-depth insights on the outcomes of all-comers undergoing PCI before and during the COVID-19 pandemic, showing that the worse in-hospital outcomes are not exclusively related to the COVID infection itself but also probably to the “collateral damage” of the COVID-19 pandemic to the entire cardiac care system.

The manuscript is well written, very interesting with a robust statistical methodology. Therefore, I do not have further revisions to propose.

Thank you for your kind commentary.

Reviewer #2: The aim of this retrospective, multicenter study was to assess outcomes of patients undergoing percutaneous coronary intervention (PCI) in Michigan during the COVID-19 pandemic in 2020 (between March and December), comparing them to outcomes in the same period of the previous year. The Authors used data from the Blue Cross Blue Shield of Michigan Cardiovascular Consortium (BMC2) PCI registry which included data from all the catheterization laboratories in Michigan. The Authors found a ~15% decrease in overall PCI volume from the pre-pandemic to the pandemic period; monthly mortality rates for primary PCI were in general higher in the pandemic period. Furthermore, patients undergoing PCI in the pre-pandemic period, those undergoing PCI in the pandemic period had a higher risk of death. This last finding was more pronounced in patients presenting with ST-segment elevation myocardial infarction (STEMI).

The topic is interesting, the analyses are elegant and the manuscript clearly written.

I have some considerations:

• Monthly mortality rates for primary PCI were in general higher in the pandemic period, a finding that was however not related to clinically significant system delays in STEMI care. As already mentioned by the Authors, one of the possible interpretations is that patients tried to avoid contacting the emergency services for fear of entering the hospitals. Indeed, it is possible that predominantly patients with very severe symptomatology contacted the emergency services or went to the hospital. In this context, it is also possible that patients waited until they had an advanced clinical situation before asking for assistance. It is known that several patients who were not infected with COVID-19 suffered complications of myocardial infarction associated with late presentation. It would be interesting to add “pain to balloon” to the data shown.

We agree with the reviewer that this is likely possibility. However we are not able ascertain this since the BMC2 database did not capture pain onset time and hence is unable to calculate the pain to balloon time.

• It is important to note also the low absolute number of STEMIs and NSTEMIs in 2020 compared to 2019: perhaps another indication that many patients stayed home and only the most symptomatic called for rescue.

This is indeed an important consideration, and it has been underlined in the discussion (page 18 and page 19). As the reviewer suggests, the lower number of ACS in 2020 might have been driven by the fact that patients tended to stay home for fear of contracting COVID-19 in the hospital: these considerations are discussed at the end of page 19.

• The authors performed outcome analyses using in-hospital mortality. Although the latter is widely used in the literature, I believe it would be more elegant and objective to use 30-day mortality, if it were available.

We agree with the reviewer that 30 day mortality can provide additional information compared with in hospital mortality. Unfortunately, the BMC2 registry collects only in-hospital data and all reported outcomes accordingly reflect in-hospital events.

---

## [Decision Letter · Decision Letter 1]

15 Aug 2022

Trends and outcomes of percutaneous coronary intervention during the COVID-19 pandemic in Michigan

PONE-D-21-35320R1

Dear Dr. Gurm,

We’re pleased to inform you that your manuscript has been judged scientifically suitable for publication and will be formally accepted for publication once it meets all outstanding technical requirements.

Kind regards,

Shukri AlSaif

Academic Editor

PLOS ONE

Additional Editor Comments (optional):

Reviewers' comments:

Reviewer's Responses to Questions

**Comments to the Author**

1. If the authors have adequately addressed your comments raised in a previous round of review and you feel that this manuscript is now acceptable for publication, you may indicate that here to bypass the “Comments to the Author” section, enter your conflict of interest statement in the “Confidential to Editor” section, and submit your "Accept" recommendation.

Reviewer #1: All comments have been addressed

Reviewer #2: All comments have been addressed

2. Is the manuscript technically sound, and do the data support the conclusions?

Reviewer #1: Yes

Reviewer #2: (No Response)

3. Has the statistical analysis been performed appropriately and rigorously? 

Reviewer #1: Yes

Reviewer #2: (No Response)

4. Have the authors made all data underlying the findings in their manuscript fully available?

Reviewer #1: Yes

Reviewer #2: (No Response)

5. Is the manuscript presented in an intelligible fashion and written in standard English?

Reviewer #1: Yes

Reviewer #2: (No Response)

6. Review Comments to the Author

Reviewer #1: Azzalini et al. report an interesting study regarding the evaluation of the influence of the COVID-19 pandemic on temporal trends and outcomes of patients undergoing percutaneous coronary intervention (PCI) during the year 2020 compared to year before (2019). The topic is of high interest due the persistence of the COVID pandemic that is bringing the world health to its knees. Despite numerous previous manuscripts depicting differences in PCI volumes and outcomes between the pandemic and the pre-pandemic era this paper covers a longer period compared to initial reports that exclusively focused on the “first wave” of the pandemic. Furthermore, it gives in-depth insights on the outcomes of all-comers undergoing PCI before and during the COVID-19 pandemic, showing that the worse in-hospital outcomes are not exclusively related to the COVID infection itself but also probably to the “collateral damage” of the COVID-19 pandemic to the entire cardiac care system.

The manuscript is well written, very interesting with a robust statistical methodology. Therefore, I do not have further revisions to propose.

Reviewer #2: (No Response)

7. PLOS authors have the option to publish the peer review history of their article (what does this mean?). If published, this will include your full peer review and any attached files.

Reviewer #1: **Yes: **Gianmarco Iannopollo

Reviewer #2: No

---

## [Editor Report · Acceptance letter]

14 Sep 2022

PONE-D-21-35320R1 

Trends and outcomes of percutaneous coronary intervention during the COVID-19 pandemic in Michigan 

Dear Dr. Gurm:

I'm pleased to inform you that your manuscript has been deemed suitable for publication in PLOS ONE. Congratulations! Your manuscript is now with our production department. 

Kind regards, 

on behalf of

Dr. Shukri AlSaif 

Academic Editor

PLOS ONE